# Panic and Trust during COVID-19: A Cross-Sectional Study on Immigrants in South Korea

**DOI:** 10.3390/healthcare9020199

**Published:** 2021-02-12

**Authors:** Myeong Sook Yoon, Israel Fisseha Feyissa, So-Won Suk

**Affiliations:** Department of Social Welfare, Jeonbuk National University, Jeonju City 54896, Korea; yoon64@jbnu.ac.kr (M.S.Y.); lovewon077@hanmail.net (S.-W.S.)

**Keywords:** COVID-19, public health, panic disorder, public trust, immigrants, global pandemic

## Abstract

In the COVID-19 pandemic, marginalized groups like migrants are disproportionately affected. As panic, fear of neglect, and mistrusting institutions in these groups are presumed to be apparent, their detachment to health services still needs to be investigated. This study comparatively analyzed the level of panic and trust between South Koreans and immigrants who are living within highly affected areas of South Korea. Mann–Whitney-U-Test and Pearson correlation showed panic is more pronounced in the Korean group while having a similar panic display pattern with the immigrants. The immigrant group appears to highly trust the Korean health system, health institutions, local media, and the local native community. Beyond conventional expectations, participant’s average panic score showed a statistically significant positive correlation with items of the trust scale, indicating a level of individual reliance amid the pandemic panic. Thus, ascertaining institutional trust and matured citizenry are identified as factors for effective public health outcomes. During such a pandemic, this study also reminded the public health needs of immigrants as secondary citizens, and presumptions of immigrants’ mistrust in such settings might not always be true.

## 1. Introduction

COVID 19 was first reported in China in November of 2019. South Korea was the first country that reported COVID-19 infection outside of China. From its first confirmed case on 20 January, up to its peak period, 1 March–31 March, the confirmed cases were 9661 with 159 deaths—That is an average of 236 confirmed cases each day [1]. South Korea used different types of strict public mobilizations to deter the spread of the virus. Although the country is mentioned among countries that averted much-anticipated infection and its consequences [2], such major pandemics are still expected to have many negative impacts that potentially induce panic on individuals [3,4]. The spread of disease and the diffusion of panic are always interrelated, both pose challenges of different kinds for public health [5].

Among other public health concerns, it is high time to assess the mental health ramifications of COVID-19. Previous studies on the psychological effects of similar instances like SARS outbreak (2002–2004) produced results like; increased levels of concern for personal and family health [6], fear of contagion, feelings of stigmatization, loneliness, boredom, anger, anxiety, and a sense of uncertainty [7]. Since the outbreak of COVID-19, the mental health of different group settings are reported to be impacted by the spread of the virus; i.e., medical, nursing staff, and other healthcare personnel [8], close contacts, people in lockdown or isolation [9], patients in clinics, families, and friends of affected people [10]. As equally important, however, the need to include the psychological effect of infectious diseases on often neglected groups still seems necessary. Reluctance to involve, understand, and include everyone as a key partner in the medical and public health response could hamper the effective management of an epidemic and increase the likelihood of social disruption [11,12,13]. In addition, inclusive public health efforts will be vital for effective containment and mitigation of the outbreak, reduce the overall number of infections, and shorten the emergency situation [14]. In this study, it was deemed important to measure the level of panic among the immigrant communities within the enclosures of highly affected areas of South Korea as opposed to the native Korean community. Although immigrants are different in types, most of them appear incapable of accessing health services in their host countries. For instance, immigrants like migrant workers or asylum seekers might face major constraints due to inadequate health insurance schemes specifically designed for them. From a public health perspective, there is a need to account for migrants in COVID-19 response and recovery efforts [15].

On the institutional level, the unique set of challenges migrants face in such a pandemic emanate from the lack of entitlement to health care, exclusion from social welfare programs, and fear of stigmatization and/or arrest and deportation [15]. The effect of COVID-19 is expected to affect low-income migrant communities disproportionately. Such an instance is reported in Buchanan and colleagues’ report [16] about a disproportionate COVID-19 infection in some of New York’s neighborhoods where migrants are overrepresented.

Access to health care services for the migrant is determined based on the legal immigration status. As is the case in many countries, migrants in irregular status or on short-term visas will be ineligible for equal access to health care and Covid-19 treatment. Such public health cracks are reported to endanger the health of migrants during the COVID-19 pandemic [17,18,19,20]. Migrants’ lack of awareness of locally recommended prevention measures due to language and cultural barriers, or adherence to culture-specific customs and practices is reported in putting migrant communities at increased risk of COVID-19 transmission [21]. Alongside the spread of the virus all over the world, xenophobic treatments towards migrants were also reported [22]. In past pandemics and during the current COVID-19 pandemic, migrants were often used as a scapegoat or stigmatized [23]. Chinese or any Asian and European migrants in countries all over the world experienced a different type of Xenophobic and racist sentiments in their host societies.

During such a pandemic, immigrants may be particularly vulnerable to the direct and indirect psychological impacts of COVID-19. Under normal circumstances, studies conducted on immigrants such as international migrant workers indicate that they have a high vulnerability to common mental disorders and a lower quality of life than local populations [24].

Therefore, immigrant living and working conditions, their language ability, language diversity in service provision, their local knowledge and networks, presence or absence of xenophobia, and their access to rights and level of inclusion in host communities, will determine the immigrants’ ability to avoid the infection, receive adequate health care and cope with the economic, social and psychological impacts of the pandemic [25].

As panic, fear of neglect, and sometimes mistrust in the host countries’ institutions are apparent reactions for the immigrants, it will only worsen the problem if their panic reactions are mismanaged or if they mistrust in institutions that are deemed to control the spread of the virus. This study then will look into how the immigrant communities reacted and cope during the peak of the spread and how they interacted with local institutions and native communities. More specifically, it is the focus of this study is to understand the position of immigrants’ panic in relation to their overall trust in the Korean public institutions and the local community while comparing them with native Koreans.

This particular study set out to sample the different types of immigrants residing in South Korea. Although the study generically differentiates the immigrant types as; Employed, Self-employed/freelancer, Student, Unemployed, and other, their socio-economic status and the level of privileges they are entitled to when it comes to health care are different from one another. The employed immigrant group represents the biggest categories of Korea’s less-skilled migrant workers and immigrants with professional visas. Migrant workers with the visit and employment visa (H-2), non-professional employment visas (E-9), and immigrants with professional visas (E series visas) are included in the study. Since 16 July 2019, this particular chunk of the immigrant group and the self-employed along with any foreigner who lived in South Korea for more than 6 months are compulsorily subscribed to national health insurance which comes with entitlement to universal health coverage. On the other hand, students, the unemployed, and other types of migrants have differed from the universal national health insurance. The other type of immigrants included in the study represents the illegal migrants with no valid immigration status.

### Panic and Institutional Trust during the COVID-19 Pandemic

In previous studies of infectious diseases such as SARS, the outweighing psychological burden was not only contracting the virus but also the fear and panic associated with it [10]. Before and during the spread of COVID-19, societies all over the world experienced panic among other various mental health complications [5,26,27].

Panic is induced by a fear of something and it is usually a “reflection of a groundless, a primitive flight response to some perceived danger” [5]. Based on the cognitive theory of panic, simple thinking about panic-related sensations and their feared effect is enough to induce panic [28]. The disruption due to COVID-19 also created what Tversky and Kahneman termed “availability bias”. According to this bias, people tend to aggrandize and fear disasters if they are highly publicized and easily comes to mind [29]. Before COVID-19 was even deemed a pandemic, constant sensationalized content on the media and frequent exposure to it was reported to create panic [5,30].

Communication and civic engagement of the public is an integral part of public health. However, due to panic people can become indifferent to public health messages [31]. Public health messages that ask for the public to comply could also negatively alarm the public [5]. In pandemic settings, it is also suggested for government institutions to utilize “non-alarmist framings of health threats because they might reduce the capacity of public health organizations to mobilize the public” [32]. As Ventrigilo and colleagues [27] mentioned in their commentary on Covid-19 and panic, panic can also push people to “respond in a rebellious way; where people may think that they know better and try to ignore government advice because it gives them some degree of control”. They also affirmed that this panic state is “likely to develop into anger where people lose faith in their respective governments where the salient and covert social contract is deemed to fail” [27].

The relationship of public panic and institutional trust within the context of the COVID-19 pandemic is recommended for further investigation [5]. Institutional trust is the confidence of the public in the actions of government institutions to do what is right and perceived to be fair [33]. This trust also depends on “the congruence between citizens” preferences—their interpretation of what is right and fair and what is unfair—and the perceived actual functioning of government [34]. Trust in the health system is “the optimistic acceptance of vulnerability in the belief that the system has best intentions” [35]. The effectiveness of restrictive policy is also highly dependent on individual trust [36]. Italy, which is ravaged by the spread of the virus, reported how misinformation about the matters of health and science within the public created panic and affected the public trust in public institutions [37]. In a “low-trust” state such as Hong Kong, public health responses to COVID-19 proved to be successful despite the low trust among the public that was expected to undermine the public health initiatives [38]. The government of Singapore used extra push to “heighten the perceived risks” of COVID-19 on the Singaporean public that overly trusted the government and underestimated risks while ignoring individual responsibilities [39].

According to the Trust–Confidence–Cooperation model [40], public trust and confidence in the health system are built on previous experience with the health system during previous emergencies and overall experience with the system. The public also will be motivated to comply with restrictive regulations when there is “value and intention-based trust” and “performance-based confidence” [41]. There is a need for studies that focus on the interaction of immigrants with their host countries’ public institutions that directly affect their daily experiences and well-being; i.e., interactions in services like social, housing, education, and health [42]. The higher trust of immigrants was reported in instances where there is a difference in the quality of governance between host and origin country or if there is a preferable institutional value in the host country [42].

In addition to studies of trust in a public institution, this particular study aims to add knowledge on how much immigrants in South Korea trust the health system, nearest health institutions, their local media, and their local community amid the crisis posed by COVID-19. Considering these public institutions and nearest local community as the direct nearest contacts to immigrants and admitting the existence of a certain level of panic during the spread of the virus, the objective of the study is to understand the severity of panic and trust level of immigrants in light of the native Korean community. It is safe to assume that not all immigrants easily access appropriate services. It is also a fact that not every immigrant in the country is adequately insured and equally positioned as the native [15]. Thus, this actual reality will only worsen if the actual potential of the country’s health service is hugely burdened. It could thus be hypothesized that an immigrant’s trust in public institutions and local communities could be negatively affected since the already overburdened public institutions will likely neglect immigrant’s needs. The study then asks the question; do the immigrants panic more and lose trust, or is it vice versa?

## 2. Materials and Methods

### 2.1. Study Design and Setting

This Observational cross-sectional study was conducted in selected locations of South Korea. These locations experienced a high spike in daily confirmed COVID-19 infection cases. These cities and provinces are Seoul, Deagu, Busan, Incheon, Gwanju, Gyeonggi province, and Gangwon province. The study included an immigrant group sample and a comparative matched native Korean sample. Immigrant participants for the study were recruited through representatives of foreign communities and an online survey was sent directly to their social media addresses. Community representatives were contacted and were advised on how to distribute the online survey. Native Korean participants were recruited by the research team after a careful matching of the socio-demographic characteristics of the immigrant participants. The study met the quality standard described in the declaration of Helsinki and the necessary Institutional Review Board approval for ethical considerations was also maintained from Jeonbuk national university (IRB file No. JBNU 2020-09-002-003).

### 2.2. Participants

The eligible participants were selected by checking their immigrant status for the immigrant group and by checking national registration Korean status for Korean participants. Physical presence in South Korea between December of 2019 to May of 2020 in the principal locations of the study was a mandatory participant selection criterion. Screening question for a past history of the Panic disorder before this particular survey was asked for both groups and those with a history of panic disorder were excluded from the survey. Participants demographic description is provided in Table 1.

Around 88% of the immigrant group participants are mainly composed of the expat community and the international student community. Self-employed, the unemployed, and other types of immigrants compose the rest 11.8%. In addition, around 77.6% of our participants are between the ages of 25–45. The Socio-demographically matched comparison group took consideration of these population characteristics and percentages.

### 2.3. Variables and Measurements

The outcome variables in this study are individual panic scores and individual trust scores. The grouping variables are immigrant status and native Korean status. The study used a customized 10 item Adult-Severity Measure for Panic Disorder and a 4 item trust survey for trust scores. Both measurements went through forward and back translation to Korean and Chinese language. Pre-testing was also employed before developing the final versions.

The original Adult Severity Measure for Panic Disorder is a 10-item measure that assesses the severity of symptoms of panic disorder in individuals age 18 and older [43]. In our study, the questions on this measurement are modified in direct relation to the panic potentially caused by COVID-19. For instance, item 1, “during the past 7 days I have felt moments of sudden terror, fear or fright, sometimes out of the blue” was changed to “during the past few weeks I have felt moments of sudden terror, fear or fright because of the spread of COVID19 in/around my city”.

Every 10 items on the measure are rated on a 5-point scale (0 = Never; 1 = occasionally; 2 = Half of the time; 3 = Most of the time, and 4 = All of the time). The total score can range from 0 to 40, with higher scores indicating greater severity of the panic disorder. Thus, scores below 1 indicate a normal level, between 1–9 indicates a mild panic disorder, 10–19 moderate level, 20–29 severe level, and 30–40 extreme level.

The 4-item survey that measures the individual’s trust focused on assessing individual trust in the public health system, institutions, media, and the local community. The questions are:1.How much do you trust the health system in the country to fully contain the spread of COVID19?2.How trustworthy is information provided about COVID19 by your nearest health institution?3.How trustworthy is your local media while disseminating COVID19 related public announcements?4.How responsive is your local community in collaborating with the instructions of health care officials?

Each item on the trust scale is rated on a 4-point scale (0 = Not at all; 1 = low level; 2 = moderately, and 3 = highly). The total score can range from 0 to 12, with higher scores indicating a higher trust in public institutions and local communities’ response to the spread of the virus.

Internal consistency was examined using Cronbach’s alpha for the 10 item panic severity measurement and the 4 item trust scale. The panic severity measurement and the trust scale on the immigrant group scored a Cronbach alpha of 0.78 and 0.71 respectively. The internal consistency of these measurements on the Korean group also scored a Cronbach’s alpha of 0.8 and 0.76 respectively.

### 2.4. Bias

The socio-demographic characteristics (age, gender, occupational status, and Location) of the immigrant and the Korean group sample are identically matched before the comparative analysis. Since normality distribution on the average trust score was violated (*p* ≤ 0.05) for independent samples T-test, we instead opted to employ a non-parametric Mann–Whitney-U-Test.

### 2.5. Study Size

As of December 2018, the total number of immigrants in South Korea is 2,367,607 [44]. Out of the total population, our target group is immigrants above the age of 18, which are 91% of the total population. Thus, a sample size of 407 immigrants with a 95% confidence level and a 4.8% margin of error was considered a fair sample size for this particular study. One-hundred Koreans, a 1:4 ratio of socio-demographically matched sample, was considered fair after a statistically significant (*p* ≤ 0.00) Welch’s test on both outcome variables.

### 2.6. Procedure

First, participants’ average severity of panic score as well as average trust level is calculated. A descriptive analysis of the whole result was put together to outline the general panic severity and the trust level of the two groups. Then, a non-parametric Mann–Whitney-U-Test was employed to determine if there is a statistically significant difference between the two groups’ mean scores of the outcome variables. Pearson correlation coefficients are also employed to determine correlations between average panic scores and individual trust scores. Test results with missing data from both groups were excluded from the study.

## 3. Results

### 3.1. Adult Panic Severity and Trust Test Results

The experience of panic between the immigrant and the native group is graphically presented in Figure 1. The immigrant group experience of panic is mostly between moderate and extreme, indicating a diagnosable level of panic experience. On the other hand, most of our Korean participant’s severity of panic falls between moderate to severe levels.

In a descriptive analysis of the high and low mean scores of the panic disorder (see Table 2); both immigrants and Koreans scored higher on item no. 7. Naturally, both groups developed a new routine, or participated only minimally in social activities, because of the fear of getting infected. Although this higher mean score in item 7 is expected to occur in legally binding national lockdown situations, it is important to note that South Korea has not implemented a national lockdown or similar type of restrictions. Both groups also scored a lower mean score on an item no. 4.

The average trust score indicate more immigrants have a higher trust on the health system, health institutions, local media, and local community. The average trust distribution is displayed in Figure 2.

As it is displayed in Figure 3, the Individual-level of trust, the mean scores of trust in the health system, nearest health institutions, and local media were higher in the immigrant group. However, the mean score of the trust level on the local community is greater in the native Korean group. The overall mean scores in both groups indicate a higher level of trust in the mentioned institutions and the local native community (M ≥ 2.0).

### 3.2. Panic: Immigrants vs. Koreans

A statistically significant difference in the items of the panic measurement scale was checked for both groups. Table 3 shows Mann–Whitney U test results for Immigrant and Korean groups’ panic severity scores.

Mann–Whitney-U test indicated that there are statistically significant differences between immigrant and Korean groups on the panic severity scale. The differences occurred in all items, except for items 4, 5, and 7. The Korean group scored significantly more highly than the immigrant groups on items 1, 2, 3, 8, and 9. Only in items 6 and 10, the immigrant group scored a significantly higher score.

### 3.3. Trust: Immigrants vs. Koreans

A statistically significant difference in the items of the trust measurement scale was checked for both groups. Mann–Whitney-U test indicated that there are statistically significant differences between immigrant and Korean groups in all items of the trust scale. According to the result, immigrants appear to have high trust scores than the natives. Table 4 shows Mann–Whitney U test results for trust scores.

Pearson correlation was used to determine associations between average panic severity score and individual trust scores. The higher individual experience panic, the higher he/she trust the health system (*r* = 0.316, *p* < 0.01). There is also a positive correlation between panic and trust in nearest health institutions (*r* = 0.370, *p* < 0.01). Trust on the local media and trust on local native community are also positively correlated with panic; (*r* = 0.427, *p* < 0.01) and (*r* = 0.243, *p* < 0.01) respectively.

## 4. Discussion

In this study, we demonstrated that there is a significant level of panic which is induced by the spread of COVID-19. There is a similarity in how panic and fear are displayed between the immigrant group and the natives. In both groups, there was a less outer physical experience of panic symptoms outlined on items 4 and 5 of the panic severity scale. These items also appear to have no statistically significant difference among the groups. Relatively high mean scores on items 4 and 5 might have indicated a panic severity that is in a dire need of immediate assistance. Within the panic severity score, it is easily understandable that both groups experienced panic that is not enough to induce physical symptoms. However, the panic experience takes the form of disrupting normal routines, prohibiting social activities, and forcing people to be extra cautious about how to avoid infections.

Based on the result of this study, South Korean institutions, as well as the local native community, are seen as highly trustworthy. With a slight difference between the immigrant and the native groups, scores of trust in both groups indicate a high level of trust in the health system, nearest health institutions, local media, and local native communities. This level of trust could potentially translate to a relatively conducive element for successful public health outcomes. It also implies the circumstances in which the country is among the high-trust state during the COVID-19 crisis. Fair and equitable sharing of health resources also mitigated further risks to the public’s health by meeting public health needs while increasing trust. During this pandemic, “Individuals with ambiguous citizenship rights, regardless of their legal status, should be offered care, to encourage them to report when they are ill and stop the spread of covid-19” [45]. Since 16 July 2019 enrollment in the National Health insurance in South Korea has become mandatory for all immigrants, except for international students. However, still, there are a group of immigrants who are still not eligible for national health insurance, like asylum seekers and undocumented immigrants. During this pandemic, the country’s universal health coverage might ease up the panic among the insured immigrant communities. Although there is no recorded case of uninsured foreigners being at the crossroad of not getting treatment or test for COVID-19, the fact of not having insurance during this pandemic might put the uninsured on edge. When it comes to testing COVID-19 and quarantine efforts, South Korea took the cost burden off anyone regardless of nationality. The country even went to the extent of encouraging illegal immigrants to come to test sites promising no legal ramifications for their illegal stay in the country.

Within the comparison, the native Koreans appear to have high panic scores on properties of the panic severity scale, while immigrants appear to score high on the trust scale. The immigrant group appears to highly trust the Korean health system, health institutions, local media, and the local native community much higher than the Korean group. The possible explanations for these levels of trust could be; the provision of almost free medical testing and medical treatment for every immigrant regardless of nationality and the immigrant’s general assessment of how the country (or local native community), responded to the COVID-19 crisis. One possible argument for the higher level of immigrant’s trust in local media could be the diverse availability of divisive and less trustworthy media outlets for Koreans compared to the few, direct, and less diverse media outlets for immigrants. People comparing their government with other governments elsewhere in how they responded to COVID-19 public health situations are reported to determine panic or feeling of safety [27]. Here it is also important to have a second look at the type of immigrants that this particular study included. Those less-skilled migrants (H-2 and E-9 visa holders), who mostly migrate from less-developed countries, will be appreciative of the universal health coverage benefits that their home countries might not provide during such pandemics. By comparison, there will be instances in which South Korea appears as a safe haven for those immigrants from less-prepared countries for the pandemic.

Timely, accurate, and transparent risk communication is vital in public health crises such as COVID-19 [27]. However, it might appear a challenge in such emergencies because it depends on how far the public willing to trust authorities more than rumors and misinformation. In such regard, South Korea’s sense of urgency, strong implementation capacity, and effective communication and public outreach strategy have distinguished the country’s approach and contributed to its effectiveness [46]. However, it is important not to ignore the fact that immigrants faced language barriers when it comes to accessing and understanding public announcements. In fact, in most cases, the department of health services in most areas of the country disseminates information mostly in Korean, which translates into a lack of information for non-Korean speakers. A large number of immigrants do not read Korean, they are dependent on assistance for directions about precautions, what to do when family members get sick, or if the government orders confinement. Although, South Korea is still learning the ropes of how to accommodate outsiders, a tailored health service for the immigrant is recommended in such public health crisis.

The highly-rated local community collaborative effort could also contribute its part by producing an effective implementation of public health campaigns. The collective actions of the public have an implied power to direct the public behavior of immigrants. The country was able to lower the number of new infections for consecutive months and the mortality rate from COVID-19 hangs at a low level. Although it is relative to other countries, this public health success is partially attributed to the law-abiding “matured citizenry” of local communities and their collaboration in the government’s extensive testing and tracing campaigns.

The overall positive correlation of panic and trust in this study is an indication of the conducive circumstance for public health directives. For instance, risk communications to establish trust in authorities have been less successful in Japan and Hong Kong [38]. From a public health point of view, the way South Korea dealt with the spread of the virus could be considered exemplary and was enough to prevent the downsides of public panic and social disruptions. Although immigrants are part of the main public, their minority status and second citizenry might have compromised their health security and trust in the institutions. However, as the result of this study indicate, the level of trust in the immigrant group is even relatively better to increase the efficiency of public health efforts that expect to mobilize everyone in the country.

Everyone in the country, including immigrants’ acclimatizing with the government’s efforts by following public guidelines such as agreeing to share personal information, wearing a mask in public, following measures, maintaining social distance, and personal hygiene indeed yielded a relatively excellent public health outcome. Additionally, the unweaving effort of the government’s top-down move to trace, test, and treat not only eased the public panic early from the beginning, but it also produced the public health security that might have translated into higher damage and casualty.

## 5. Conclusions

Our study provided a comparative outlook on how immigrant communities within highly affected areas of South Korea reacted to the spread of the virus within highly affected areas of South Korea. The study also indicated the role of trust within effective public health outcomes. Based on conventional understanding, when panic increases individual is susceptible to decreases trust in public institutions and starts to develop self-reliance [11]. Such self-reliance is potentially detrimental to collective public health initiatives. However, in this study, the positive relationship of panic and trust could only explain the immigrant’s reliance on the institutions and local community amid collectively shared mass panic.

The result of the study also pinpointed the importance of ascertaining trust and matured citizenry to avert public panic and greater mental health outcomes in such a global pandemic. The study also reminded the public health needs of immigrants as secondary citizens. Immigrants as secondary citizens of any country are not only vulnerable groups in such pandemic, but they are also systematically detached from services and access to public health services. Due to such instances; Panic, fear of neglect, or mistrust of host countries’ institutions are presumably apparent among immigrant communities. However, such presumptions might not always be true, at least based on the results of this study.

## 6. Limitations of the Study

The limitation of this study is the convenience of the sampling we used. There was difficulty in including more samples of relatively vulnerable immigrant groups such as undocumented immigrants and asylum seekers during the process of the study. Based on their circumstances, samples from these types of individuals might have provided a different view. This inevitable lack of random or representative sampling strategy may result in several sample-specific differences that limit the generalizability of the findings.

## Figures and Tables

**Figure 1 healthcare-09-00199-f001:**
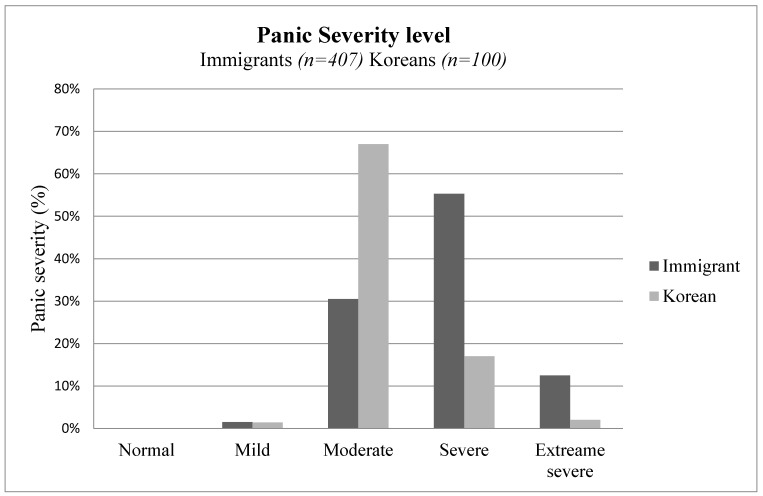
Panic severity level in immigrants (*n* = 407) and Koreans (*n* = 100).

**Figure 2 healthcare-09-00199-f002:**
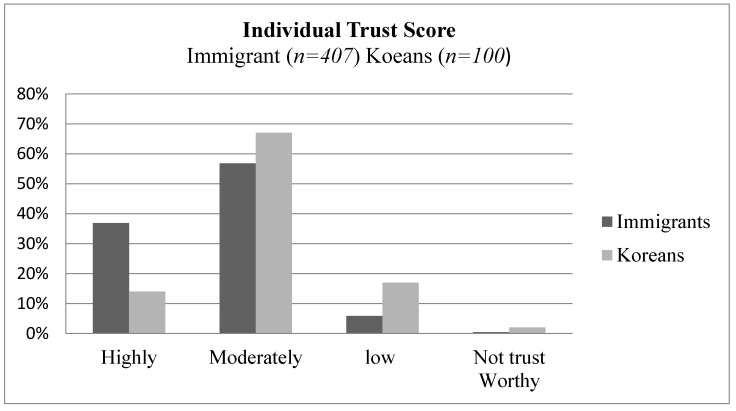
Average trust score (*n* = 407) and Koreans (*n* = 100).

**Figure 3 healthcare-09-00199-f003:**
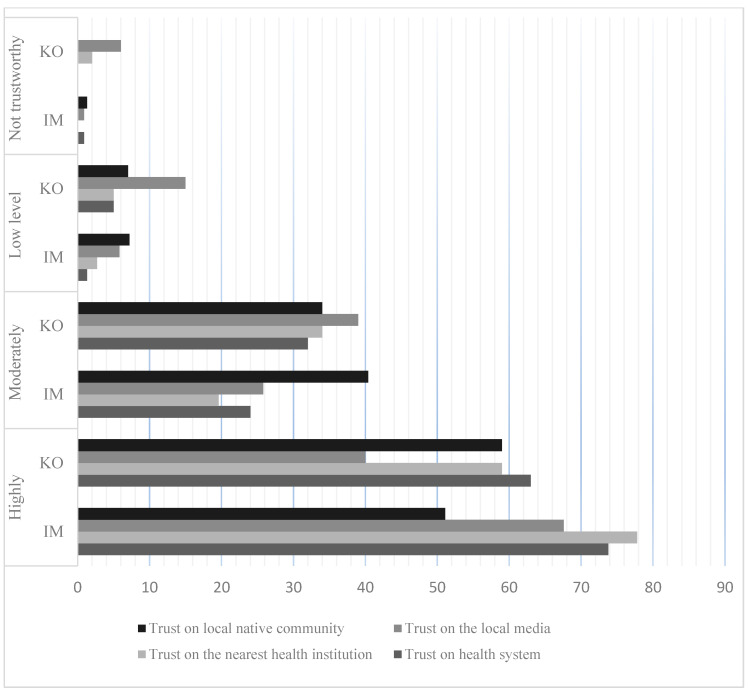
Individual-level trust result (*n* = 407) and Koreans (*n* = 100).

**Table 1 healthcare-09-00199-t001:** Participants description.

		Immigrants in South Korea (*n* = 407)		Koreans (*n =* 100)	
Item		N	%	N	%
Sex	Male	207	49.1	49	49
	Female	200	50.4	51	51
Age	18–25	35	8.6	9	9
	26–35	225	55.3	56	56
	36–45	92	22.6	23	23
	46–55	42	10.3	10	10
	Above 56	13	9.1	7	7
Occupation	Employed	190	46.7	47	47
	Self-employed/Freelancer	20	4.9	5	5
	Student	168	41.3	41	41
	Unemployed	15	3.7	4	4
	other	13	3.2	3	3
Locations	Seoul city	181	44.4	44	44
	Daegu city	52	12.7	13	13
	Busan city	42	10.3	10	10
	Incheon city	22	5.4	6	6
	Gwanju city	31	7.6	8	8
	Gyeonggi province	49	12	12	12
	Gangwon province	30	7.3	7	7
Nationality	China	181	44.4		
	USA	43	10.5		
	Nigeria	23	5.6		
	Ethiopia	23	5.6		
	Canada	20	4.9		
	Bangladesh	18	4.4		
	Others *	99	24.3		

* origin of other nationalities: Afghanistan, Australia, Belgium, Botswana, Bulgaria, Burundi, Cambodia, Cameroon, Chile, Croatia, Denmark, DR Congo, Ecuador, Egypt, Finland, France, Gabon, Germany, Ghana, India, Iran, Ireland, Kenya, Liberia, Malaysia, Mali, Mexico, Mongolia, Morocco, Nepal, Netherlands, New Zealand, Niger, Pakistan, Philippines, Poland, Russian Federation, Rwanda, Singapore, South Africa, Spain, Tanzania, UK, Uzbekistan, Vietnam, Zimbabwe.

**Table 2 healthcare-09-00199-t002:** Descriptive analysis/panic disorder mean scores.

Item		Immigrants (*n* = 407)		Koreans (*n* = 100)	
		M	SD	M	SD
1	I felt moments of sudden terror, fear or fright because of the spread of COVID19 in/around my city	1.15	0.98	1.8	1.3
2	I felt anxious, worried, or nervous about getting infected with COVID19	1.39	1.06	2.0	1.35
3	I had thoughts of losing control, dying, going crazy, or other bad things happening because of COVID19 in my area.	0.69	0.99	1.0	1.1
4	I felt a racing heart, sweaty, trouble breathing, faint, or shaky because of the spread of COVID19 in/around my city	0.24	0.56	0.39	0.9
5	I felt tense muscles, felt on edge or restless, or had trouble relaxing or trouble to sleep because of the spread of COVID19 in/around my city	0.53	0.8	0.6	1.0
6	I avoided, or did not approach or enter, situations that reminded you of the spread of COVID19	1.5	1.3	1.1	1.3
7	I developed a new routine, or participated only minimally in social activities, because of the fear of getting infected by COVID19	2.56	1.2	2.50	1.23
8	I spent a lot of time preparing for situations in which I might avoid the troubles of getting infected by COVID19	1.81	1.3	2.1	0.84
9	I distracted myself to avoid thinking about the current situations of COVID19 in my area.	1.46	1.28	0.8	0.9
10	I needed help to cope with the fear of COVID19 spreading rapidly (e.g., alcohol or medication, superstitious objects, prayer, other people)	0.89	1.18	0.56	0.94

**Table 3 healthcare-09-00199-t003:** Mann–Whitney U-Test on panic severity scale.

	Item	Group	*n*	Mean Rank	Sum of	U	*p*
**1**	I felt moments of sudden terror, fear or fright because of the spread of COVID19 in/around my city	IM	405	237.16	96,049	13,834	0.000 ***
		K	100	317.16	31,716		
**2**	I felt anxious, worried, or nervous about getting infected with COVID19	IM	407	241	98,121.50	15,093.5	0.000 ***
		K	100	306.57	30,656.50		
**3**	I had thoughts of losing control, dying, going crazy, or other bad things happening because of COVID19 in my area.	IM	407	245.12	99,764	16,736	0.002 **
		K	100	290.14	29,014		
**4**	I felt a racing heart, sweaty, trouble breathing, faint, or shaky because of the spread of COVID19 in/around my city	IM	407	251.68	102,433.50	19,405.5	0.291
		K	100	263.45	26,344.50		
**5**	I felt tense muscles, felt on edge or restless, or had trouble relaxing or trouble to sleep because of the spread of COVID19	IM	407	254.17	103,447	20,281	0.951
		K	100	253.31	25,331		
**6**	I avoided, or did not approach or enter, situations that reminded me of the spread of COVID19	IM	407	262.15	106,697	17,031	0.009 **
		K	100	220.81	22,081		
**7**	I developed a new routine, or participated only minimally in social activities, because of the fear of getting infected by COVID19	IM	407	255.48	103,978.50	19,749.5	0.635
		K	100	248.00	24,799.50		
**8**	I spent a lot of time preparing for situations in which I might avoid the troubles of getting infected by COVID19	IM	407	244.81	99,636	16,608	0.005 **
		K	99	289.24	28,635		
**9**	I distracted myself to avoid thinking about the current situations of COVID19 in my area.	IM	407	220.85	89,884	6856	0.000 ***
		K	100	388.94	38,894		
**10**	I needed help to cope with the fear of COVID19 spreading rapidly (e.g., alcohol or medication, superstitious objects, prayer, other people)	IM	407	261.88	106,586.50	17,141.5	0.007 **
		K	100	221.92	22,191.50		

** *p* < 0.01; *** *p* < 0.001.

**Table 4 healthcare-09-00199-t004:** Mann–Whitney U-Test on Trust scale.

	Item	Group	*n*	Mean Rank	Sum of	U	*p*
**1**	Trust on health system	IM	407	294.45	119,840	3888	0.000 ***
		K	100	89.38	8938		
**2**	Trust on the nearest health institution	IM	407	296.20	120,554	3174	0.000 ***
		K	100	82.24	8224		
**3**	Trust on the local media	IM	407	293.99	119,654	4074	0.000 ***
		K	100	91.24	9124		
**4**	Trust on local native community	IM	403	284.65	114,713.5	6992.5	0.000 ***
		K	100	120.43	12,042.5		

*** *p* < 0.001.

## Data Availability

The data presented in this study are available on request from the corresponding author. The data are not publicly available due to ethical reasons.

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
