# Peer review of "Panic and Trust during COVID-19: A Cross-Sectional Study on Immigrants in South Korea"

_healthcare, 2021, doi:10.3390/healthcare9020199_

Round 1

Reviewer 1 Report

The study aimed to measure and investigate panic and trust differences between immigrants and native people in South Korea during the COVID-19 Pandemic.  The authors first introduced how the outbreak of COVID-19 may have affected people’s mental health, generating panic and reducing trust’s level towards local bodies among the general population. The authors furthermore introduced how immigrants are often set aside and appear incapable of accessing health care in their host countries; explaining why there is a need to account for migrants in COVID-19 response and recovery effort.

The article follows a coherent flow without any major break. I have listed below a series of points that requires however the authors attention. In particular some of these represent major points that needs to be accounted and must be revised. The others represent suggestions to the authors and some small typo errors.

I suggest therefore to accept the article only if and after these major points are accounted for and revised:

Major concern:

  • Demographic details about participants and covariates
    • Demographic information about the immigrants seems to be too generic. Not all the immigrants are the same and not all of them have the access to the same level of health care and level of social communities. It is not clear to which group of immigrants this study is referring to. Some more information about the different level of access to health care in different groups of immigrants may be needed in the introduction,
    • Other possible covariants in the study:
      • Immigrants culture and lifestyle, some cultures are usually more open to integrate themselves with native people.
      • Age in the country may represent an important factor, as the above it may affect how much people are integrated in the local culture and with the local population, affecting therefore their judgment of the public institutions and local communities.
      • Among the employed group a different level of income may impact the individual judgment of local institutions and communities. Were these matched as well among the two groups?

  • Line 202
    • Could you please provide more info about the decision to use a 4 point Likert scale?
    • Employing a 4 item likert scale does not allow the participants to take a neutral position and force them to force an opinion. The scale aims to measure trust of public health system, institutions, media, and the local community, it seems naturally therefore that some participants may have a neutral opinion, while they are forced to take a position employing a 4 point Likert scale.
  • Line 281
    • Furthermore, each item of the trust questionnaire seems to measure the trust of participants in different local bodies health system, health institution, local media and native community. It is not clear whether these are going to be analysed together or individually. If they are going to be measured individually as it seems on line 281, the use of a single item for an individual local body seems not appropriate and I would personally suggest expanding the number of items. Furthermore, if each item measures the trust in a different local body internal consistency would not be an adequate measure of reliability

  • Line 263
    • It is not clear why the inferential analysis was run on each item of the Panic Questionnaire instead of calculating the average total score for the questionnaire for each participant before running the inferential analysis. This must be clarified.

Lower concern:

  • Abstract:

……. to highly trust Korean health system (U= 3888, P = < .001), health institutions (U= 3174, P = < .001), local media (U= 16736, P = < .001), and the local native community (U= 15 6992, P = < .001). Individual average panic score also have a statistically significant positive corrrelation with items of the trust scale. Ascertaining institutional trust and matured citizenry are identified as factors for effective public health outcome. During such pandemic, this study also reminded 18 the public health needs of immigrants as secondary citizens and presumptions of immigrants mistrust in such settings might not always be true.

I suggest removing statistics data from the abstract replacing them with text. Statistics results sometimes are hard to grasp when reading an abstract and would best fit in the result section.

  • Line 24

COVID 19 was first reported in China by November of 2019. South Korea was the first country that reported COVID-19 infection outside of China. From its first confirmed  case on January, up to its peak period, March 1st - March 31, the confirmed cases were 9661 with 159 deaths

Add Citations backing the information about the dates.

  • Line 43

 Reluctance to involve, understand, and include everyone as a key partner in the medical and public-health response could  hamper effective management of an epidemic and increase the likelihood of social disruption’ [10].

The cited paper seems not fully appropriate, despite discussing the argument, it does it in light of the event of a bioterrorist attack, which even if it may lead to similar consequences and social disruption, it does not resemble the case of COVID-19. The paper however cites other works discussing how previous epidemics events were treated in the past. I suggest adding some of these citation in addition to the already present one.

  • Line 49 to 73.

As this part lay down the rational of the study, it would be nice to further extend it including more detail of the phenomenon.

  • Line 88

 However, due to panic people can become indifferent to public health messages.

Missing Citations

  • Line 92

As Antonio and colleagues [17] mentioned in….

Citation is wrong, Antonio is the author’s name not surname.

  • Line 106

A country such as Italy, which is ravaged by the spread of the virus, reported how misinformation about the matters of health and science within the public created panic and affected trust on public institutions [26].

Subject and/or verb is missing, please rephrase

  • Line 119

In studies of trust of immigrants in public institutions, it is recommended to include studies that focus on the interaction of immigrants with public institutions that directly affect the daily experiences of migrants and their well-being such as social, housing, education and health services [31].

This sentence is not clear, I suggest rephrasing.

  • Line 125

As an addition to studies of trust on public institution, this particular study is aiming to add knowledge on how much immigrants in South Korea trust; the health system, nearest health institutions, their local media, and their local community amid the crisis posed by COVID-19.

The use of “;” in this sentence appears to be wrong.

  • Line 224

First, the Individual test….

What test Is this line referring to? “Individual test” written in capital letters hint to the name of a specific test, this is however never mentioned. Please rephrase this sentence

  • Line 235

As it is graphically presented in Figure 1, there is a difference in the experience of  panic between the immigrant and the native group. The immigrant group experience of  panic is mostly between moderate and extreme, indicating a diagnosable level of panic experience. On the other hand, most of our Korean participant's severity of panic falls between moderate to severe level.

It should not be stated that there is a difference between the two groups based on the graph. Furthermore, please state what is the variable on the y axis.

  • Line 247

Between  the two groups however, the mean score of the immigrants appears slightly greater than the natives indicating immigrants experienced slightly greater disruption to normal routines.

The authors in this sentence seem to draw conclusions based on descriptive stats. Please rephrase

  • Line 261

A graph reporting data in Table 3 would be much easier to understand for the reader, allowing to visually catch the differences between the two groups.

  • Line 303

groups indicate a high level of trust in; the health system, nearest health institutions, local media, and local native communities.

The use of “;” in this sentence appears to be wrong.

Reviewer 2 Report

The present manuscript entitled "Panic and Trust during COVID-19: A cross sectional study on immigrants in South Korea" is a study with interest, but its methodological and ethical weaknesses make it, from my point of view, not publishable. The modifications to be made and considered are listed below point by point:

1. In tables and figures, the uppercase N, which is used for the entire sample, and the lowercase n are not used correctly.

2. In line 208 modify "Dec".

3. In the methodology section there is no review of the ethical aspect. The study must have been approved by an ethics committee that assigns a reference number. The lack of this approval makes this manuscript non-publishable. Like nothing is mentioned about the information sheet and informed consent for the study participants.

4. The discussion section is not discussed with other studies.

5. Conclusions should be concise, not a summary.

Round 2

Reviewer 2 Report

The authors in the manuscript have corrected and modified the indicated point by point. My recommendation is to accept it.
